REGISTERED REPORT PROTOCOL

# Profiles of competence development in upper secondary education and their predictors

**Micha-Josia Freund** [1]*, **Ilka Wolter** [1], **Kathrin Lockl** [1], **Timo Gnambs** [1,2]

1 Leibniz Institute for Educational Trajectories, Bamberg, Germany, 2 Johannes Kepler University Linz, Linz, Austria

☉ These authors contributed equally to this work.
* Micha.Freund@lifbi.de

**Data Availability Statement:** As we work with secondary data, this dataset cannot simply be

## Abstract

This registered report protocol elaborates on the theory, methods, and material of a study to identify latent profiles of competence development in reading and mathematics among German students in upper secondary education. It is expected that generalized (reading and mathematical competence develop similarly) and specialized (one of the domains develops faster) competence profiles will be identified. Moreover, it is hypothesized that students' domain-specific interest and educational history will predict membership of these latent profiles as these factors influence the students' learning environments. For this study, we will use data from the German *National Educational Panel Study*, including students from ninth grade in secondary schools (expected $N = 14,500$). These students were tracked across six years and provided competence assessments on three occasions. The latent profiles based on the students' reading and mathematical competences will be identified using latent growth mixture modeling. If different types of profiles can be identified, multinomial regression will be utilized to analyze whether the likelihood of belonging to a certain competence development profile is influenced by students' domain-specific interest or educational history. As this protocol is submitted before any analyses were conducted, it will provide neither results nor conclusions.

## 1 Introduction

Language and mathematical competences significantly impact academic and professional success. Basic language competences (including reading competence) are at the core of learning and communicating [1], while basic mathematical competence (or mathematical literacy) is defined by the Organization for Economic Co-Operation and Development (OECD) ([2], p.15) as the ability "to make well-founded judgements and to use and engage with mathematics in ways that meet the needs of that individual's life as a constructive, concerned and reflective citizen". Both competence domains are thus basic skills necessary for everyday life, which is why both reading and mathematical competences are often analyzed in educational research.

Students in secondary education display consistent development in reading and mathematical competences with a reduced growth rate towards the end of compulsory education [3–5].

published with the final article. However, the data is accessible to scientists fulfilling all necessary requirements of the national education panel study (NEPS) once they have concluded a Data Use Agreement with the Leibniz Institute for Educational Trajectories (https://www.neps-data. de/Data-Center/Data-Access). Once a scientist has access to the data, the data will be available on the NEPS-homepage (https://www.neps-data.de/ Mainpage).

**Funding:** The publication of this article was funded by the Open Access Fund of the Leibniz Association.

**Competing interests:** The authors have declared that no competing interests exist.

The two domains are highly correlated in cross-sectional data in both early lower [3] and early upper secondary education [6]. Previous research on the relationship between the development in reading and mathematical competences demonstrated substantial correlations between the change trajectories in both domains throughout secondary education [3, 7]. However, at the end of mandatory education, research on domain-specific competence development and especially on the relationship between the two domains through a longitudinal perspective is scarce.

Against this background, this paper aims to analyze the longitudinal trajectories of mathematics and reading competence by identifying profiles of competence development of students in Germany at the beginning of upper secondary education, commencing in grade 9 until age 21/22. We expect these profiles to be either generalized profiles of competence development (similar development in both reading and mathematical competence) or specialized profiles of competence development (a higher development in either domain). In a previous study with students at the beginning of lower secondary school in Germany (grades five to nine), we were unable to confirm specialized profiles of competence development in those domains [8]. However, based on the manifold options which the German educational system offers in upper secondary school, a higher level of specialization is expected in this period of schooling. If the expected profiles of competence development are found, potential predictors of profile-membership will also be analyzed.

## 1.1 Individual's characteristics as determinants of competence development in reading and mathematics

Certain student characteristics can influence the development of reading and mathematical competence development of all students. Some of these explain the high correlation between mathematical and reading competences. In this context, research has shown that underlying abilities such as working memory [9–11] and reasoning ability [12] impact both domains. For example, several studies discovered working memory to be substantially correlated to both language and mathematical competences [13–15]. In a recent meta-analysis by Peng et al. [16], working memory and reasoning abilities together accounted for over 50% of the variance in the relation between language and mathematics. Additionally, the correlation between mathematical and reading competences can also be traced back to the fact that general language and reading competences are important for learning in general but also for acquiring mathematical knowledge and solving mathematical problems [1, 16–18].

Previous research has shown that, in addition to the internal factors, socio-demographic characteristics of the students impact their competence development. Mathematical and reading competences are highly correlated to the socioeconomic status of students' parents even before elementary education [19] and in development through secondary education [20]. As a summary of studies by Shin and colleagues [3] shows, the gap between students from high and low socioeconomic backgrounds can be shown to increase, decrease, or stagnate depending on the model, tests, and sample, making specific longitudinal effects of social background on profiles of competence development hard to work out. Nonetheless, the socio-economic background can be seen as a determinant of both competence domains simultaneously, further indicating generalized profiles of competence development.

On the other hand, differences in reading and mathematical abilities were confirmed for male and female students. Cross-sectional studies in this field depict that, on average, boys have higher mathematical and lower reading competence in grade 9 compared to girls [21]. These inter-individual (between-student) differences imply potential intra-individual (within-student) differences between the domains at least cross-sectionally. Longitudinal development

of gender-differences is less obvious, with studies showing differences decreasing [22] or stagnating [23] in secondary education. Thus, while the effect of gender on cross-sectional competence differences seems quite clear, longitudinal effects are difficult to predict.

Socio-demographic characteristics are not the only individual determinants of competence development implying potential specialization. Affective-motivational (e.g., motivation [24, 25], interest [26]) or socio-cognitive (e.g., self-concept [26]) determinants also have clear impacts on competence development in the domains and may differ between the domains for some students. As can be shown with the example of interest, research finds differences between the interests of students in academic domains [26, 27]. This should, in turn, be highly related to school-related or leisure time activities in these domains [28], leading to higher or lower achievement in the domains. Ehrtmann, Wolter, and Hannover [27] showed that many sixth-grade students' interest in German and mathematics (as well as further vocational interest domains) might be classified as generalized high or low, but some students can be classified into a profile with high interest in mathematics and low interest in German, as well as a profile with high interest in German and low interest in mathematics. Due to the aforementioned connection of interest and investment, we expect that students are more likely to be found in a profile of specialized competence development if they are distinctively more interested in one of the domains than the other. The existence of both generalized and specialized profiles of interest overall implies the existence of these profiles in competence development as well.

## 1.2 Context characteristics as determinants of competence development in reading and mathematics

Finally, the learning context should also play a role in competence trajectories. That is, competence development in both domains should be affected by the characteristics of teaching in the classroom and the type of school a student attends [29] but also by students' choices during their educational career. Differences in the development of mathematical and reading competences in upper secondary school might be enforced by specific characteristics of the German educational system. With the end of lower secondary school and compulsory schooling after the ninth grade, the German system offers multiple pathways in either further general education towards a university entrance certificate or vocational training and according exams [30]. Compared to lower secondary education, where students study the same courses within their school types, in upper secondary education and vocational training, they have more options to specialize in a domain. Students in the highest school track Gymnasium, for example, can choose between basic and advanced courses [31], which also determines parts of their exams at the end of schooling.

Students in vocational training [32] are already selecting their occupations and should therefore also more likely show specialized competence development. It is plausible to assume that their competence profiles specialize throughout their vocational training due to the focus of their apprenticeships on job-specific skills that might consist of predominantly language (e.g., reading), or mathematics-related tasks. Similarly, after finishing upper secondary education with a university entrance certificate, students entering university can decide on a university course, which might include predominantly language- or reading-related competences (e.g., arts or language studies) or mathematical competences (e.g., science, technology, engineering, or mathematics [33]). We would thus expect that students in specific vocational training or university study programs are also more likely to be specialized in their competence development in reading and mathematics than students not in specific vocational training or university courses. Overall, the existence of more pathways and courses in upper secondary education further strengthens the argument that some specialized profiles of competence development should be found.

## 2 Hypotheses

Against this background, we expect to identify not only a generalized profile of competence development with a similar trajectory for mathematical and reading competence but also specialized profiles of competence development at the beginning of upper secondary education. More specifically, we expect two specialized profiles of competence development, which are differentiated into a predominantly mathematical competence and a reading competence profile.

*Hypothesis 1*: *There are one generalized and two specialized profiles of competence development.*

Learning environments of students after grade 9 might influence their likelihood of belonging to either specialized or generalized profiles of competence development. Specialized interest can be interpreted as a higher likelihood of specializing the leisure time use to acquire either mathematical or reading competences leading to higher competences in the specific domain. Similarly, students might focus more on one domain through further education. Vocational education after grade 9 and higher education after grade 12 prepare for a career in a specific sector or job. As that sector or job might demand a higher competence level in either reading or mathematics, a high specificity of vocational or higher education could lead to a higher likelihood of ending up with a specialized profile of competence development.

*Hypothesis 2*: *Students with interests predominantly in one domain, reading or mathematics, are more likely to specialize in that domain than students with an unspecialized interest.*

*Hypothesis 3*: *Students who choose an occupation or a university program in a STEM field in school more likely belong to a specialized profile in mathematics than in reading. Corresponding to this, students who choose an occupation or a university program identified as reading-centered are more likely to belong to a specialized profile in reading than in mathematics.*

## 3 Materials and methods

### 3.1 Sample

The study will use data from a sub-sample (starting cohort Grade 9) of the German *National Educational Panel Study* (NEPS [34]), which examined representative samples of students from secondary schools across their educational careers. All NEPS-data is publicly available to scientists meeting all requirements and after the conclusion of a Data Use Agreement (https://www.neps-data.de/Data-Center/Data-Access/Data-Use-Agreements) with the Leibniz Institute for Educational Trajectories. In NEPS, students were sampled in a multi-stage stratified cluster design [35]. The present study (expected $N = 14,500$), focuses on students who were initially tested in mathematics and reading in grade 9 (expected age $M = 15$) and, subsequently, received competence tests in mathematics and reading at three-year intervals. The sample is expected to include about 49% females and about 27% with a migration background. According to the publicly available information (https://www.neps-data.de/Data-Center/Data-and-Documentation/Starting-Cohort-Grade-9/Documentation), about 37% of students are expected to attend Gymnasium or the Gymnasium branch of comprehensive school.

### 3.2 Knowledge of data

The lead author has never worked with this dataset. All theories and hypotheses, as well as details on the methodological approach, are based on a thorough literature review and prior research on other samples of the NEPS, including a currently unpublished paper with a similar

aim in a mutually exclusive dataset. The co-authors have previously worked with the dataset, albeit on topics unrelated to the present research. All publications using NEPS data published by the authoring team can be found at https://www.neps-data.de/Project-Overview/Publications (filtering for starting cohort 4). Furthermore, the co-authors also contributed to some unpublished papers, which used the present dataset. However, none of the authors conducted analyses pertaining to this preregistration, including identifying profiles of competence development or identifying profiles across multiple domains. The authors thus have no knowledge of the results of this study yet. All information used in this protocol (e.g., regarding the sample description) was derived from the documentation available online (https://www.neps-data.de/Data-Center/Data-and-Documentation/Starting-Cohort-Grade-9/Documentation).

## 3.3 Instruments

In the ninth grade, mathematical and reading competences were measured in a class-context, whereas later assessments were conducted individually in the students' private homes by trained test supervisors. Information on students' backgrounds, as well as on predictor variables, was taken from a questionnaire answered by the students. Details for the test and all questionnaire variables provided in the dataset can be found on the NEPS homepage (https://www.neps-data.de/Data-Center/Data-and-Documentation/Starting-Cohort-Grade-9).

**3.3.1 Mathematical competence.** Mathematical competence tests with items from four content areas and six cognitive components were specifically developed for use in the NEPS [36]. The mathematical tests at the beginning of grades nine, twelve, and three years after grade 12 consisted of 22, 21, and 21 items, respectively [37–39]. They include simple and complex multiple-choice items, as well as short constructed responses. Item response theory was used for scaling the tests [40]. Weighted maximum likelihood estimates (WLE) [41] and linking across grades with the help of overlapping items were used to attain student proficiencies [42]. Reliabilities of the WLEs in the three grades were .81, .77, and .75, respectively. To compare the competences in the two domains, the WLEs will be standardized according to the mean and standard deviation in grade 9.

**3.3.2 Reading competence.** Reading competence tests in NEPS are constructed according to a theoretical framework with three cognitive requirements and five text types [43]. These tests were administered at the end of grade nine, beginning of grade twelve, and three years after grade twelve. They consist of 31, 28, and either 23 or 27 items, respectively. The number of items in the last test differs because of different difficulty-tiered booklets depending on prior reading competence levels [44–46]. The different tests were placed on a common scale using an anchor-test design [42] to allow for valid longitudinal change analyses. Reliabilities of the WLEs for reading competence were .81, .80, and .77, respectively. The WLEs will be standardized according to the mean and standard deviation in grade 9.

**3.3.3 Additional variables.** To test hypotheses two and three, we will include further variables in our analyses. To measure students' interest in academic domains (mathematics and German) in NEPS, a scale was adapted from Baumert and colleagues [47]. Students were asked four items per domain in grade 9 on their interest in spending time on mathematics and literature. The four questions for each domain were then turned into a scale. After z-standardizing the scales, the difference scores between the interests in the two domains will be calculated and used as a metric scale to indicate specialization of interest.

Additionally, to analyze whether students spent significant time in reading or mathematically specialized education, all episodes of schooling, training, or studying that were at least six months long will be considered. Each of these episodes will be classified as either language specialized, mathematically specialized, or generalized. Vocational trainings that are defined as

STEM (science, technology, engineering, or mathematics) occupations by the Federal Employment Agency of Germany [48] and university programs in the fields of mathematics, natural sciences, and engineering [33] will be coded as mathematically-specialized. Vocational trainings in the area of law, print-media, archives, and libraries, as well as university programs in the fields of language and cultural studies, will be coded as reading-specialized. All other episodes will be coded as generalized (or unspecialized) episodes. Once every episode has been coded, students will be checked whether they spent significant time (at least six months) in only one of the two specialized areas (thus being specialized) or in both or in none (being generalized).

In addition to these predictor variables, several additional variables are necessary that will be used for imputation in addition to competence and predictor variables. These variables include unique identifiers for the student and their school. Gender is already available in the dataset with 7,853 male and 7,692 female students in wave 1. The age of students will be calculated in months by subtracting the month and year of the test in grade nine from their birth month and year (most students are born between 1994 and 1996, indicating roughly a mean age of 15.5 at wave 1). The highest occupational prestige of the parents (defined as a parent questioned in a questionnaire and their partner) using the International Socio-Economic Index (ISEI) of Occupational Status [49] and the highest number of years in education of the parents using the CASMIN (Comparative Analysis of Social Mobility in Industrial Nations) classification [50] will be used as social background characteristics of students. To create a variable accounting for the type of school in grade 9, all schools leading to university entrance qualification (Gymnasium, & the Gymnasium branch of Comprehensive schools–ca 5,510 students in wave 1) will be differentiated from all other types of schools (ca. 9,383 students in wave 1).

Migration background will be re-coded to compare students with a first- or second-generation migration background (student or at least one parent born in another country—about 4,262 students in wave 1) to all other students (11,481). A scale of interaction language in different contexts will be created by taking the average of six variables on a students' interaction language: with their mother, with their father, with their siblings, with their best friend, at the schoolyard, and of the parents with each other. The self-concept of students will also be considered using variables adapted from Kunter and colleagues [51]. This test includes 10 items on the self-concept of students in the school classes on German and mathematics (five items each). Finally, a test on reasoning abilities [52] will be included in the dataset. The original test included 12 items, which tests if students can identify the right figural element to complete a given figural sequence. An overview of all variables can be found in Table 1.

## 3.4. Statistical analyses

An overview over the statistical process, including the used datasets and variables at each step, can be found in Fig 1.

**3.4.1 Latent change analyses.** Longitudinal competence development will be analyzed using linear latent growth models (LGM) [53]. The basic model will provide information about the initial competence (intercept) and development (slope) of all students. Specifically, a dual-process LGM (with two slopes and two intercepts) will be specified to take into account both mathematical and reading competences. This model will be estimated in Mplus using a maximum likelihood estimator [54] with 4,000 initial stage starts and 1,000 final stage optimizations. Then, latent growth mixture modeling (LGMM) [55, 56] will identify the different profiles of competence development. As the focus of this study is on the development of students (and not initial competence) our model will only use the mean LGM slopes of

**Table 1. List and description of all variables used in this study.**

| Variable | | Necessary transformation | Range of values |
|---|---|---|---|
| *Competence* | | | |
| Reading competence | Grade 9 | z-standardization | -∞ to +∞ |
| | Grade 12 | | |
| | Grade 12 + 3 years | | |
| Mathematical competence | Grade 9 | | |
| | Grade 12 | | |
| | Grade 12 + 3 years | | |
| *Predictors* | | | |
| Specialization of interest | | Creation of scale | -∞ to +∞ |
| Specialization of education | | Creation of scale | -1, 0, 1 |
| *Controls* | | | |
| Gender of the student | | - | 0, 1 |
| Migration background | | Dichotomization | 0, 1 |
| Type of school in grade 9 | | Dichotomization | 0, 1 |
| Highest CASMIN of parents | | Creation of scale | 9 to 16 |
| Highest ISEI of parents | | Creation of scale | 16 to 90 |
| Interaction language of students | | Creation of scale | 0 to 3 |
| *Additional auxiliary variables (for imputation)* | | | |
| Age of students at first testing | | Calculation | 0 to +∞ |
| Self-concept in German | | - | 1 to 4 |
| Self-concept in mathematics | | - | 1 to 4 |
| Reasoning ability of students | | - | 0 to 16 |

mathematical and reading competences to allocate profiles of competence development. As such, the intercepts in both domains will be constrained across all profiles.

**3.4.2 Dealing with missing values.** To account for the dropouts in the data of NEPS, we will use a multiple imputation approach [57]. We will impute missing values 30 times using predictive mean matching in the Stata-package ICE [58]. For imputation, we will use age, type of school in grade 9, interaction language of the students, migration background, reasoning abilities, the domain-specific self-concept in German and mathematics classes, the highest ISEI and the highest CASMIN of the parents in addition to the competence tests in mathematics and reading for each grade and the aforementioned predictor variables (gender, specialization of further educational paths, and specialization of interest in mathematics or reading).

**3.4.3 Model selection.** To identify the optimal number of profiles, we will fit different LGMMs with 1 to 10 classes. Then, we will exclude models with profiles including less than 5% of the students. Smaller profiles are likely difficult to replicate and seem to have negligible practical relevance. In a next step, the model with the best fit will be chosen using the Bayesian Information Criterion (BIC) [59] and the Lo-Mendel-Rubin Likelihood Test (LMRT) [60, 61]. The model with the lowest BIC and a significant LMRT can be interpreted as the model with the best fit. A significant ($\alpha$ = .05) LMRT indicates that a model with $k$ profiles provides a better fit than a model with $k$-1 profiles. All criteria for model selection are summarized in Table 2.

**3.4.4 Interpretation of profiles.** The basic LGM will act as a baseline to interpret the profiles of the other models. We will take the sum of both slopes in the LGM and divide it by 4. This value will serve as a criterion for interpretation. If the difference between the two slopes in a profile is greater than this criterion, students differ more in their development between

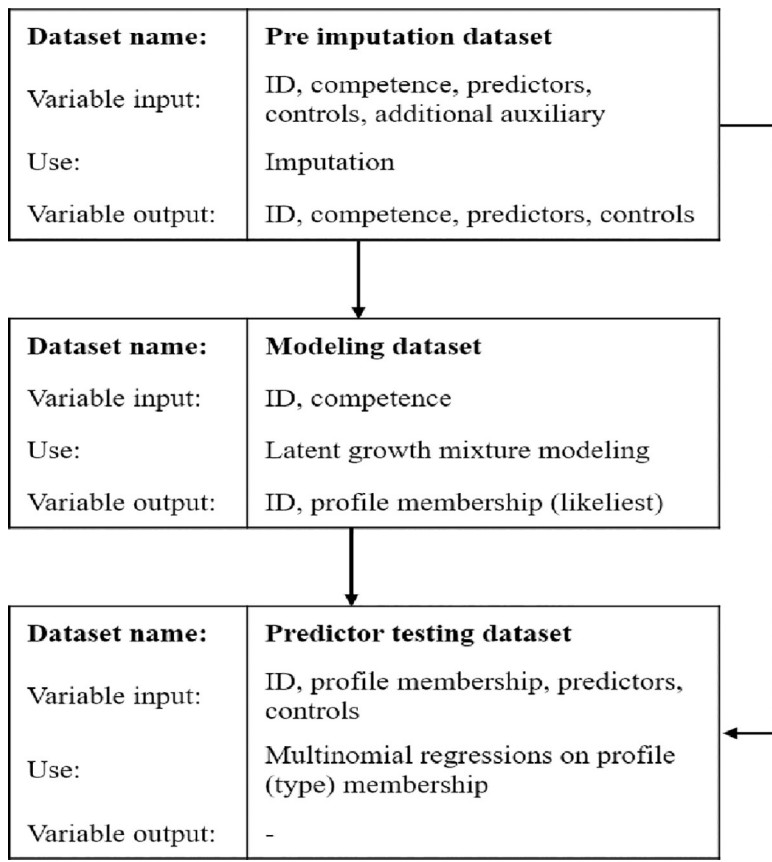

**Fig 1. The three statistical steps, necessary datasets and variables.**

the domains than the average student develops within half a year. Profiles with a higher difference will be interpreted as specialized profiles of competence development while profiles with a lower difference are interpreted as generalized. All profiles should fit into one of these three types of profiles, as only this difference between the slopes (and not the absolute level of slopes or intercepts) is relevant for profile interpretation.

However, it is possible, that this classification results in several profiles of the same type. For example, it is conceivable that two specialized profiles appear that simply differ in their degree of specialization (i.e., the amount of difference in slopes). However, differences within profile types are not the focus of the present study. Therefore, for the prediction analyses, if more than one profile of a type is identified, these profiles will be merged into a single profile type. For example, two generalized profiles, two mathematically specialized profiles and one reading specialized profile would be condensed into three profiles, each containing all original latent profiles of their type.

**Table 2. Criteria for model selection.**

| Name | Type of criterion | Decision making process |
|---|---|---|
| Profile size | Exclusion criterion | Profile size of every profile at least 5% |
| BIC | Fit index | Lowest BIC indicates best fit |
| LMRT | Fit index | Last significant LMRT indicates best fit |

**3.4.5 Testing predictors.** If we can identify both generalized and specialized profiles, we will be able to test the influence of the predictors on the likelihood of belonging to each class via a three-step approach [62]. In this approach, the most likely class and the measurement errors for each student (calculated in step one in the LGMM) are saved as manifest variables (step two). The effect of the predictors on the likelihood of class-membership is then tested via multinomial regression (step three). In this regression, both predictors and several additional control-variables will be used. The control variables are gender, school type in grade nine, socioeconomic background, migration background, and interaction language of the students. As an inference criterion for the effect of interest and educational pathways, we will use a significance level of 1%.

## Author Contributions

**Conceptualization:** Micha-Josia Freund, Ilka Wolter, Kathrin Lockl, Timo Gnambs.

**Writing – original draft:** Micha-Josia Freund.

**Writing – review & editing:** Micha-Josia Freund, Ilka Wolter, Kathrin Lockl, Timo Gnambs.

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
