## [Decision Letter · Decision Letter 0]

12 Nov 2020

PONE-D-20-23037

Profiles of competence development in upper secondary education and their predictors

PLOS ONE

Dear Dr. Freund,

Thank you for submitting your manuscript to PLOS ONE. After careful consideration, we feel that it has merit but does not fully meet PLOS ONE’s publication criteria as it currently stands. Therefore, we invite you to submit a revised version of the manuscript that addresses the points raised during the review process.

Both reviewers acknowledged the strong potential of the proposal and that it will be a valuable contribution to the existing literature. However, reviewer 1 also indicated several smaller issues which I think you can easily address, and they would make your registered report more robust and transparent. I'm looking forward to the revised manuscript.

We look forward to receiving your revised manuscript.

Kind regards,

Vitomir Kovanovic, Ph.D.

Academic Editor

PLOS ONE

Journal Requirements:

Reviewers' comments:

Reviewer's Responses to Questions

**Comments to the Author**

1. Does the manuscript provide a valid rationale for the proposed study, with clearly identified and justified research questions?

Reviewer #1: Yes

Reviewer #2: Yes

2. Is the protocol technically sound and planned in a manner that will lead to a meaningful outcome and allow testing the stated hypotheses?

Reviewer #1: Partly

Reviewer #2: Yes

3. Is the methodology feasible and described in sufficient detail to allow the work to be replicable?

Reviewer #1: Yes

Reviewer #2: Yes

4. Have the authors described where all data underlying the findings will be made available when the study is complete?

Reviewer #1: Yes

Reviewer #2: Yes

5. Is the manuscript presented in an intelligible fashion and written in standard English?

Reviewer #1: Yes

Reviewer #2: Yes

6. Review Comments to the Author

You may also provide optional suggestions and comments to authors that they might find helpful in planning their study.

Reviewer #1: Section 3.3

I would recommend including a table of the data (if possible) such as listing attribute names and values that will feed your model. This would allow the reader to gain a better understanding of what data you included and the relationship it has with your hypothesis and methodological approach.

Section 3.4.3

Include a table (if possible) that shows the processes of model evaluation, i.e. best fit statistics, number of classes and associated model. This will help the reader how you determined which model was the best choice.

My only concern is how do you know for sure that a generalised and a specialised profile of competency will be the only viable latent profiles? What if the model yields a competency based profile that is completely unexpected? For example, a latent profile that shares both characteristics of a generalised and specialised competency or perhaps something entirely different? How would you address a model that recommends having 4 or 5 latent profiles as the best fit? It did not seem clear to me that the data used and methodology can and only yield either a generalised or specialised profile of competency.

Regardless, I believe this research has a lot of potential.

Reviewer #2: This seems like a straightforward study for which the origin of the data, as well as the purpose is very clear. Maybe the discussion and analysis of all presented background variables in relation with the latent profiles is a bit too ambitious but I nevertheless deem them feasible. I'm looking very much forward to the results.

7. PLOS authors have the option to publish the peer review history of their article (what does this mean?). If published, this will include your full peer review and any attached files.

Reviewer #1: No

Reviewer #2: **Yes: **Philipp Sonnleitner

---

## [Author Response · Author response to Decision Letter 0]

18 Dec 2020

Dear reviewers, 

thank you for your very helpful and rather positive reviews. We have tried to adress all proposed and expected changes. This included adding two tables as reviewer 1 proposed and adding a deeper explanation into the interpretation of profiles throughout the paper. We hope these changes fulfil your expectations. 

Specific comments are adressed in the "Response to reviewers" file. Thank you for your help in improving this paper. 

With regards, 

Micha-Josia Freund

---

## [Decision Letter · Decision Letter 1]

11 Jan 2021

Profiles of competence development in upper secondary education and their predictors

PONE-D-20-23037R1

Dear Dr. Freund,

We’re pleased to inform you that your manuscript has been judged scientifically suitable for publication and will be formally accepted for publication once it meets all outstanding technical requirements.

Kind regards,

Vitomir Kovanovic, Ph.D.

Academic Editor

PLOS ONE

Additional Editor Comments (optional):

The reviewer 1, who had some small issues identified in the previous round, indicated that all the concerns were adequately addressed, so no need for any further changes. 

Reviewers' comments:

Reviewer's Responses to Questions

**Comments to the Author**

1. Does the manuscript provide a valid rationale for the proposed study, with clearly identified and justified research questions?

Reviewer #1: Yes

2. Is the protocol technically sound and planned in a manner that will lead to a meaningful outcome and allow testing the stated hypotheses?

Reviewer #1: Yes

3. Is the methodology feasible and described in sufficient detail to allow the work to be replicable?

Reviewer #1: Yes

4. Have the authors described where all data underlying the findings will be made available when the study is complete?

Reviewer #1: Yes

5. Is the manuscript presented in an intelligible fashion and written in standard English?

Reviewer #1: Yes

6. Review Comments to the Author

You may also provide optional suggestions and comments to authors that they might find helpful in planning their study.

Reviewer #1: Thank you for taking the time and addressing the comments for Manuscript Number PONE-D-20-23037R1. The revised manuscript changes are significant, and I believe that all comments were adequately addressed to be accepted for PLOS ONE.

7. PLOS authors have the option to publish the peer review history of their article (what does this mean?). If published, this will include your full peer review and any attached files.

Reviewer #1: No

---

## [Editor Report · Acceptance letter]

21 Jan 2021

PONE-D-20-23037R1 

Profiles of competence development in upper secondary education
and their predictors 

Dear Dr. Freund:

I'm pleased to inform you that your manuscript has been deemed suitable for publication in PLOS ONE. Congratulations! Your manuscript is now with our production department. 

Kind regards, 

on behalf of

Dr. Vitomir Kovanovic 

Academic Editor

PLOS ONE